# Triple Eagle: Simple, Fast and Practical Budget-Feasible Mechanisms

**Kai Han**\*
School of Computer Science and Technology
Soochow University, P.R.China
`hankai@suda.edu.cn`

**You Wu**
School of Computer Science and Technology
Soochow University, P.R.China
`20235227129@stu.suda.edu.cn`

**He Huang**
School of Computer Science and Technology
Soochow University, P.R.China
`huangh@suda.edu.cn`

**Shuang Cui**
School of Computer Science and Technology
University of Science and Technology of China
`lakers@mail.ustc.edu.cn`

## Abstract

We revisit the classical problem of designing Budget-Feasible Mechanisms (BFMs) for submodular valuation functions, which has been extensively studied since the seminal paper of Singer [FOCS'10] due to its wide applications in crowdsourcing and social marketing. We propose TripleEagle, a novel algorithmic framework for designing BFMs, based on which we present several simple yet effective BFMs that achieve better approximation ratios than the state-of-the-art work for both monotone and non-monotone submodular valuation functions. Moreover, our BFMs are the first in the literature to achieve linear complexities while ensuring obvious strategyproofness, making them more practical than the previous BFMs. We conduct extensive experiments to evaluate the empirical performance of our BFMs, and the experimental results strongly demonstrate the efficiency and effectiveness of our approach.

## 1 Introduction

In the celebrated Influence Maximization (IM) problem (e.g., [19, 21, 27]) an advertiser needs to select some "seed users" in a social network, such that the influence spread caused by "Word-of-Mouth" propagation is maximized. However, the users may want some rewards for serving as seeds and also behave strategically to maximize their utilities. So the advertiser has to pay the users under some budget $B$ and ensure their *truthfulness*, besides maximizing the influence spread [36]. Another similar problem arises in crowdsourcing markets (e.g., Amazon's Mechanical Turk [1]), where a crowdsourcing task owner needs to hire some strategic workers to perform a crowdsourcing task and pay them under a budget, with the goal of maximizing the crowdsourcing revenue [4, 25, 37, 38].

Motivated by the above applications, Singer [35] initiates the studies on *budget-feasible mechanism design*, where a buyer needs to select a set $S$ of strategic sellers with private costs from a ground set $\mathcal{N}$ with $|\mathcal{N}| = n$ and pay them under a budget $B$, such that the valuation $f(S)$ is maximized while ensuring that the sellers report their private costs truthfully. Singer [35] considers a *monotone and submodular* valuation function $f(\cdot)$ and adopts a *value oracle* model where $f(X)$ can be obtained for all $X \subseteq \mathcal{N}$. Following Singer [35]'s work, a lot of BFMs have been proposed; almost all of the theoretical studies in this line put their emphasis on improving the approximation ratios of the proposed BFMs (as elaborated in Sec. 1.1), and many studies have applied BFMs to various

---

\*Kai Han is the corresponding author (Email: hankai@suda.edu.cn).

37th Conference on Neural Information Processing Systems (NeurIPS 2023).

application scenarios such as mobile crowdsensing, federated learning, experimental design, social advertising, vehicle sharing, data pricing, team formation, cellular traffic planning, and so on.

## 1.1 Main Results of the Related Work

There exist BFMs concentrating on different valuation functions including (monotone or non-monotone) submodular functions, additive, XOS and subadditive functions. We review them separately in the following.

**BFMs for monotone subomdular valuation functions:** Singer [35] proposes a 117.7-approximation BFM. Chen et al. [13] improve the work of [35] by proposing a 7.91-approximation BFM, and they also propose a lower bound of 2 (resp. $\sqrt{2}+1$) for the approximation ratio of any randomized (resp. deterministic) BFM for submodular valuation functions. Jalaly and Tardos [23, 24] further propose an improved 5-approximation BFM. It is noted that all the above mentioned BFMs are randomized and incur a polynomial number (at least $\Omega(n^2)$) of value oracle queries. The studies of [23, 24, 35] also propose BFMs with exponential time complexity, among which the randomized BFM proposed by [23, 24] achieve the best ratio of 4. Recently, Balkanski et al. [9] propose the first deterministic BFM with polynomial running time, which achieves a ratio of 4.75 by using $\mathcal{O}(n^2 \log n)$ value oracle queries, and they also show a lower bound of 4.5 for the ratio of their proposed BFM.

Observing the large number of participants in some BFM applications such as crowdsourcing, some studies [4, 23, 24] have also considered a "large market" model where the additional assumptions of $\frac{c_{max}}{B} \approx 0$ or $\frac{v_{max}}{\text{OPT}} \approx 0$ are made, with $c_{max}$ (resp. $v_{max}$) denoting the maximum cost (resp. valuation) of any single element. Under these assumptions, Anari et al. [4] and Jalaly and Tardos [23, 24] show that there exist 3-approximation and 2.58-approximation BFMs using $\mathcal{O}(n^2)$ and $\mathcal{O}(n^6)$ value oracle queries, respectively. However, their time complexity still can be too high for large markets, and the additional assumptions mentioned above may not always hold. For example, in the influence maximization problem, the advertiser's budget could be relatively small even though the social network has billion-scale, so $\frac{c_{max}}{B} \approx 0$ may not be true. Therefore, in this work we will consider the original BFM model of [35].

**BFMs for non-monotone subomdular valuation functions:** When the valuation function is non-monotone and submodular, Amanatidis et al. [2, 3] propose a 505-approximation BFM leveraging the continuous greedy algorithm in [28], while Balkanski et al. [9] achieve a 64-approximation under $\mathcal{O}(n^2 \log n)$ time complexity. Very recently, Huang et al. [22] propose a randomized BFM with an improved approximation ratio of $(3 + \sqrt{5})^2 \approx 27.4$ under $\mathcal{O}(n \log n)$ time complexity.

**BFMs for other valuation functions:** There also exist BFMs for additive valuation functions [4, 13, 18, 35], among which Gravin et al. [18] achieve the best-known approximation ratios of 2 and 3 for randomized and deterministic BFMs, respectively. For subaddtive and XOS valuation functions, the work of [9, 10] proposes BFMs using *demand oracle queries*, as it is known that no BFMs with polynomial number of value oracle queries can achieve any ratio better than $n^{0.5-\epsilon}$ for these valuation functions [35].

## 1.2 Limitations of Prior Art and Our Contributions

Despite the great progress on BFM design in a dozen years as described above, the existing BFMs for submodular valuations still suffer from several major deficiencies. First, all of them have at least $\Omega(n^2)$ query complexity, which can be impractical for large markets. Second, it is unclear whether their approximation ratios can be further improved (both for randomized and deterministic BFMs). Third, from a practical point of view, their algorithmic frameworks for achieving truthfulness still have some serious drawbacks. To understand these, we explain their techniques in the following.

The BFMs in [4, 10, 13, 23, 24, 35] have used *sealed-bid auctions* roughly described as follows: the sellers first report their costs and then are sorted according to the non-increasing order of their *densities* (i.e., the ratios of their *marginal gains* to costs), and finally the mechanism selects some winners with the largest densities and ensures truthfulness using Myerson's lemma [32]. However, sealed-bid auctions have been criticized for lacking practicability [6, 9, 26, 30], because it is hard for the players to verify truthfulness and to trust the auctioneer, resulting in strategic behaviours.

To address the issues of sealed-bid auctions, there is growing interest in designing "simple mechanisms" (e.g., [7, 15, 30, 31, 33]). The seminal work of Milgrom and Segal [31] proposes a new class of auctions dubbed *clock auctions*, where each player can be offered by multiple descending prices

and remains active only if the player accepts the last offered price, and eventually the mechanism selects some active players as the winners and pay them the last offered prices. Milgrom and Segal [31] indicate that clock auctions satisfy *obvious strategyproofness* and several other properties (e.g., group strategyproofness, transparency, and unconditional winner privacy) not possessed by sealed-bid auctions. Inspired by [31], Balkanski et al. [9] propose a BFM using clock auction with the following design: they make $\mathcal{O}(\log_2 n)$ guesses on OPT using a "doubling trick" similar to that in online learning theory [34], and create $\mathcal{O}(\log_2 n)$ candidate solutions without violating $B$ by offering each active seller a new price for each guessed OPT, and the final winners are selected from the best candidate solutions. The clock auction BFM in [22] adopts a similar framework as [9], but uses random sampling to handle non-monotone submodular valuation functions.

Although clock auctions are considered to be more practical than sealed-bid auctions [9], we argue that they sometimes still lack practicability, as enquiring a player with too many descending prices may be time-consuming and cause the effect that the player loses patience and quits the auction early. We hereby propose the concept of *pricing complexity* of clock auctions:

**Definition 1** *The (worst-case) pricing complexity of a clock auction is the maximum number of prices it offers to any player participating in the auction.*

It can be seen from above that the BFMs in [9] have $\mathcal{O}(\log_2 n)$ pricing complexity, which could still be large for impatient sellers. In fact, Balkanski et al. [9] raise an open question of "whether there exist even simpler families of budget feasible mechanisms with which one can obtain constant approximations mechanisms", and prove that *posted-price mechanisms* are insufficient for deriving a constant ratio. Note that posted-price mechanisms (e.g., [12]) are perhaps the simplest form of truthful mechanisms; these mechanisms show each player one "take-it-or-leave-it" price and hence have $\mathcal{O}(1)$ pricing complexity. The main difference between posted-price mechanisms and clock auctions is about how they deal with any seller $u$ who accept the offered price: it is mandatory to select such a seller $u$ as a winner in posted-pricing mechanisms, while clock auctions may not follow this rule (this implies that clock auctions have the additional power of use "side observations" on the other sellers' behavior to decide whether $u$ should be a winner). However, from a practical perspective, clock auctions with low pricing complexities are almost as simple as posted-pricing mechanisms, because both of them can be efficiently implemented and can be easily understood by the players for ensuring truthfulness.

In this paper, we address all the deficiencies of the existing BFMs mentioned above by presenting TripleEagle, a novel clock auction framework that brings us several simple, fast, effective, and practical BFMs. Specifically, we show that:

- For monotone submodular functions, there exist a randomized BFM and a deterministic BFM with the approximation ratios of $\frac{\sqrt{13}+5}{2} \approx 4.3$ and $2 + \sqrt{6} \approx 4.45$, respectively; both of them use at most $\mathcal{O}(n)$ value oracle queries. This improves the best-known ratio of 4.75 in [9] (using $\mathcal{O}(n^2 \log n)$ value oracle queries) for this long-standing problem.

- For non-monotone submodular functions, there exists a randomized BFM with an approximation ratio of 12, using at most $\mathcal{O}(n)$ value oracle queries. This improves the best-known ratio of $(3 + \sqrt{5})^2 \approx 27.4$ of the randomized BFM proposed in [22] using $(n \log n)$ value oracle queries.

- Under the assumption of [9, 13] that each seller's cost is no more than $B$, our TripleEagle BFMs offer each seller only one price; without this assumption, at most one additional price query is needed in total. Therefore, our BFMs only have $\mathcal{O}(1)$ pricing complexity, which provides a confirmative answer to the open question of Balkanski et al. [9] mentioned above, because TripleEagle is virtually as simple as a posted-pricing mechanism.

- We conduct extensive experiments using the applications of influence maximization and crowdsourcing. The experimental results show that, compared to the state-of-the-art BFMs proposed in [9, 22–24], our BFMs are faster in orders of magnitude, while achieving significantly better valuations of the winning sellers.

Compared to [9], our TripleEagle framework has an essentially different design roughly explained as follows. Instead of offering multiple descending prices to each seller through blindly guessing OPT as in [9], we adopt a pricing rule inspired by the simple yet fundamental *law of supply and demand* in economics [17]. Specifically, TripleEagle processes the sellers in an arbitrary order and calculates the

price offered to a new "supplier" (i.e., an unprocessed seller) based on the "current supply" (i.e., the valuation of already processed sellers who accepted the offered prices). Therefore, the offered prices in TripleEagle tend to decrease when there are more sellers. Finally, TripleEagle tries to return a set of sellers with the maximum ratios of marginal gains to prices under the budget $B$. Due to this novel design, TripleEagle only needs one scan over the sellers and offer one price to each seller, while still achieving a better approximation ratio than [9].

As far as we know, our work is the first attempt on reducing both the time complexity and pricing complexity of BFMs. We analogize a low pricing complexity to the "Triple Eagle" hole score in golf [2], and believe that our ideas have the potential to be applied to other clock auction problems.

## 2   Preliminaries

Following [35], we assume that each seller in a ground set $\mathcal{N}$ has a private cost $c(u) \geq 0$ and there is a submodular valuation function defined on the power set of $\mathcal{N}$ satisfying $\forall X \subseteq \mathcal{N} : f(X) \geq 0$ and

$$\forall X, Y \subseteq \mathcal{N} : f(X) + f(Y) \geq f(X \cup Y) + f(X \cap Y) \tag{1}$$

We say that $f(\cdot)$ is monotone if $\forall X \subseteq Y \subseteq \mathcal{N} : f(X) \leq f(Y)$, otherwise it is non-monotone. For simplicity, we call $f(X \mid Y) \triangleq f(X \cup Y) - f(Y)$ as the marginal gain of $X$ with respect to $Y$ for all $X, Y \subseteq \mathcal{N}$, and we use $c(X)$ and $p(X)$ ($\forall X \subseteq \mathcal{N}$) as shorthand for $\sum_{u \in X} c(u)$ and $\sum_{u \in X} p(u)$, respectively. We will consider clock auctions where each seller $u \in \mathcal{N}$ can be offered by descending prices that are computed using public information, and we use $p(u)$ to denote the last price offered to $u$ at any time. When a clock auction terminates, a set $S \subseteq \mathcal{N}$ can be selected as winners only if $p(S) \leq B$ and each seller $u \in S$ has never rejected any price offered to $u$. The approximation ratio of $S$ is defined as $\text{OPT}/f(S)$, where $\text{OPT} = f(O)$ and $O$ is an optimal solution to the problem $\max\{f(X) : X \subseteq \mathcal{N} \wedge c(X) \leq B\}$. We will also consider randomized clock auctions: such an auction is a deterministic clock auction for every realization of all its internal random choices, and it clearly satisfies *universal truthfulness* [22]. For the simplicity of description, we adopt the assumption made in [9, 13] that $\forall v \in \mathcal{N} : c(v) \leq B$, and we will show how to remove this assumption later. In our algorithms, if the elements in a set $X$ are sequentially inserted into $X$, listed as $\{u_1, \cdots, u_q\}$ according to the order that they are inserted, then we call the subset $Y = \{u_s, \cdots, u_t\} \subseteq X (\forall s, t : 1 \leq s \leq t \leq q)$ as a *regular subset* of $A$, and also call $Y$ a *suffix* (resp. *prefix*) of $X$ if $t = q$ (resp. $s = 1$). Throughout the paper, we use $u^*$ to denote a seller in $\mathcal{N}$ who has the maximum valuation, i.e., $u^* = \arg\max_{u \in \mathcal{N}} f(u)$.

## 3   Randomized BFM for Monotone Submodular Valuations

In this section, we consider a monotone submodular valuation function $f(\cdot)$ and provide a randomized BFM using the TripleEagle framework, as shown by Algorithm 1 (i.e., the TripleEagleRan algorithm). Algorithm 1 takes as input a parameter $\alpha > 1$ for setting a "reserve price" that will be explained shortly. In Line 1, Algorithm 1 calls the LSDPricing procedure (i.e., Algorithm 2) to process all sellers in $\mathcal{N}$ except $u^*$; each of these sellers is offered one price by Algorithm 2 and then Algorithm 2 returns a set $A$ containing the sellers who accept their prices. After that, if $f(A) \geq f(u^*)$, then Algorithm 1 calls Algorithm 2 again to process $u^*$ and finally returns a suffix of $A$ respecting the budget constraint (Lines 3-4). Otherwise, Algorithm 1 makes a random decision of returning either $u^*$ or $T$ (Lines 6-13), where $T$ is a superset of $A$ and is got by trying to include $u^*$ by offering $u^*$ a "best effort" price of $B - p(A)$.

It can be seen that Algorithm 2 offers each seller $u$ a price $\frac{Bf(u|A)}{f(A) + \alpha f(u^*)}$. This pricing rule is inspired by the fundamental Law of Supply and Demand (LSD) in economics [17], as intuitively explained in the following. First, note that $A$ contains all the processed sellers who accept the offered prices, so $f(A)$ can be regarded as the current "supply" in the market, thus the price offered to a new "supplier" (i.e., an unprocessed user $u$) should decrease with the increment of $f(A)$. Second, the marginal gain $f(u \mid A)$ represents how much value the user $u$ can contribute given the current supply $f(A)$, so the price offered to $u$ should increase with $f(u \mid A)$. Third, the factor $\alpha f(u^*)$ guarantees that no user can be paid a price larger than $B/\alpha$, which can be regarded as a "reserve price" of the buyer. Without this reserve price, the winner set could have poor value due to the overpayment to some sellers.

---

[2]e.g., a "hole-in-one" on a par-five, see https://en.wikipedia.org/wiki/Par_(score)

---

**Algorithm 1:** TripleEagleRan($\alpha$)

---

1   $A \leftarrow \mathsf{LSDPricing}(\emptyset, \mathcal{N} \setminus \{u^*\}, \alpha f(u^*))$;
2   **if** $f(A) \geq f(u^*)$ **then**
3     |   $A \leftarrow \mathsf{LSDPricing}(A, \{u^*\}, \alpha f(u^*))$;
4     |   $S \leftarrow$ the largest suffix of $A$ satisfying $\sum_{u \in S} p(u) \leq B$;
5   **else**
6     |   Sample a random number $Z$ from the uniform distribution $\mathcal{U}[0, 1]$;
7     |   **if** $Z \leq \frac{\alpha}{\alpha+2}$ **then**
8     |     |   $p(u^*) \leftarrow B; S \leftarrow \{u^*\}$
9     |   **else**
10     |     |   Set $p(u^*) = B - \sum_{u \in A} p(u)$ and show $u^*$ the price $p(u^*)$;
11     |     |   **if** $u^*$ *accepts* $p(u^*)$ **then** $T \leftarrow A \cup \{u^*\}$ ;
12     |     |   **else** $T \leftarrow A$ ;
13     |     |   $S \leftarrow T$;
14   **return** $S$

---

---

**Algorithm 2:** LSDPricing($A, C, \eta$)

---

1   **foreach** $u \in C$ **do**
2     |   $p(u) \leftarrow \frac{B \cdot f(u|A)}{f(A)+\eta}$; Show $u$ the price $p(u)$;
3     |   **if** $u$ *accepts* $p(u)$ **then** $A \leftarrow A \cup \{u\}$ ;
4   **return** $A$

---

### 3.1   Performance Analysis of TripleEagleRan

It is evident that Algorithm 1 provides each seller only one price and incurs $2n$ value oracle queries ($n$ queries for finding $u^*$). So we only analyze its approximation ratio. Let us consider the set $A$ at the moment that Algorithm 1 finishes. For convenience, we use $A_u$ to denote the set of elements already in $A$ at the moment right before $u$ ($\forall u \in \mathcal{N}$) is processed. We first introduce Lemmas 1-2, which can be proved based on submodularity and the observation that each seller who accepts the offered price must have a sufficiently large density (and vice versa) due to the pricing rule of Algorithm 2.

**Lemma 1** *For any regular subset $X = \{u_s, \cdots, u_t\}$ of $A$ returned by Algorithm 2, we have*

$$f(X) \geq \sum_{v \in X} f(v \mid A_v) \geq p(X) \cdot \frac{f(A_{u_s}) + \alpha f(u^*)}{B}$$

**Lemma 2** *After Line 3 of Algorithm 1 is executed, we have* $\mathrm{OPT} \leq 2f(A) + \alpha f(u^*)$.

With Lemmas 1-2, we can bound the approximation ratio of Algorithm 1 if $S$ is generated by Line 4, as shown by Lemma 3. Intuitively, if $p(A) \leq B$, then we have $f(A) \geq f(u^*)$ due to Line 3 and hence we can use Lemma 2 to get $\mathrm{OPT} \leq (2+\alpha)f(S)$ directly; if $p(A) > B$, then $p(S)$ is also sufficiently large, which implies $f(S)$ is large, because $S$ is a suffix of $A$ and hence contains the sellers in $A$ with the largest ratios of marginal gains to prices due to the pricing rule of Algorithm 2.

**Lemma 3** *For any $\alpha \in (2, \sqrt{3} + 1)$, the set $S$ generated by Line 4 of Algorithm 1 satisfies*

$$\mathrm{OPT} \leq \max\left(3 + \frac{2\alpha + 1}{\alpha^2 - 1}, 2 + \alpha\right) \cdot f(S)$$

**Proof**:     Due to Line 2, we must have $f(u^*) \leq f(A)$ when Line 4 of Algorithm 1 is executed. Therefore, if $p(A) \leq B$, then we have $S = A$ and hence

$$\mathrm{OPT} \leq 2f(A) + \alpha f(u^*) \leq (2 + \alpha)f(S) \tag{2}$$

due to Lemma 2. In the following, we consider the case of $p(A) > B$. Let $H$ be the smallest suffix of $A$ satisfying $p(H) > B$, so $H \setminus S$ contains exactly one element (denoted by $w$) which is added into $A$ before the elements in $S$. Using Lemma 2 and submodularity, we get

$$
\begin{aligned}
\text{OPT} &\leq 2f(A) + \alpha f(u^*) \leq 2f(S \cup \{w\}) + 2f(A \setminus H) + \alpha f(u^*) \\
&\leq 2f(S) + 2f(A \setminus H) + (2 + \alpha)f(u^*)
\end{aligned}
\tag{3}
$$

Suppose that $f(A \setminus H) = \sigma f(u^*)$, $p(w) = \varrho B$ (and hence $p(S) > (1 - \varrho)B$) for certain $\sigma > 0$ and $\varrho \in [0,1]$. According to Lemma 1, we have

$$
f(u^*) \geq f(w \mid A \setminus H) \geq \varrho B \cdot \frac{f(A \setminus H) + \alpha f(u^*)}{B} \geq \varrho(\sigma + \alpha)f(u^*)
\tag{4}
$$

$$
f(S) \geq (1 - \varrho)B \cdot \frac{f(A \setminus H) + f(w \mid A \setminus H) + \alpha f(u^*)}{B} \geq (1 - \varrho^2)(\sigma + \alpha)f(u^*)
\tag{5}
$$

Let $x = \sigma + \alpha$. We have $\varrho \leq \frac{1}{x}$ and $f(S) \geq \frac{x^2 - 1}{x}f(u^*)$ due to Eqns. (4)-(5). Combining these with Eqn. (3), we get

$$
\begin{aligned}
\text{OPT} &\leq 2f(S) + (2 + 2\sigma + \alpha)f(u^*) \leq 2f(S) + \frac{2 + 2x - \alpha}{x - 1/x}f(S) \\
&= \left(4 + \frac{2 - \alpha + 2/x}{x - 1/x}\right)f(S)
\end{aligned}
\tag{6}
$$

Note that $x = \sigma + \alpha \geq \alpha > 2$. The R.H.S. of Eqn (6) is no more than $4f(S)$ when $x \geq \frac{2}{\alpha - 2}$ or $\alpha \geq \frac{2}{\alpha - 2}$. When $\alpha < \frac{2}{\alpha - 2}$ (i.e., $\alpha < \sqrt{3} + 1$) and $x \in [\alpha, \frac{2}{\alpha - 2})$, the R.H.S. of Eqn (6) is maximized when $x = \alpha$. Therefore, for any $\alpha \in (2, \sqrt{3} + 1)$ we have

$$
\text{OPT} \leq \left(4 + \frac{2\alpha - \alpha^2 + 2}{\alpha^2 - 1}\right)f(S) = \left(3 + \frac{2\alpha + 1}{\alpha^2 - 1}\right)f(S)
\tag{7}
$$

Combining Eqn. (2) and Eqn. (7) completes the proof. $\square$

Next, we bound the approximation ratio of Algorithm 1 when $S$ is generated by Lines 6-13 (using randomization), as shown by Lemma 4:

**Lemma 4** *For any $\alpha > 2$, if Algorithm 1 returns $S$ that is generated by Lines 6-13, then we have*

$$
\text{OPT} \leq (2 + \alpha) \cdot \mathbb{E}[f(S)]
$$

**Proof**: In this case, we must have $f(A) < f(u^*)$ after Line 1 is executed. This implies $p(A) < B$ because otherwise we have $f(A) \geq \alpha f(u^*)$ due to Lemma 1, contradicting $f(A) < f(u^*)$ when $\alpha > 1$. We prove that the set $T$ generated by Lines 10-13 of Algorithm 1 must satisfy $\text{OPT} \leq 2f(T) + \alpha f(u^*)$ according to the following discussion:

1. $u^* \notin O$: In this case, we can use similar reasoning as Lemma 2 to get

$$
\text{OPT} \leq 2f(A) + \alpha f(u^*) \leq 2f(T) + \alpha f(u^*)
$$

where the second inequality is due to $A \subseteq T$ according to Lines 11-12.

2. $u^* \in O$ and $u^*$ accepts the price in Line 11: In this case, we have $T = A \cup \{u^*\}$ and hence can also use similar reasoning as Lemma 2 to get

$$
\text{OPT} - f(T) \leq \sum_{u \in O \setminus T} f(u \mid A) \leq f(A) + \alpha f(u^*) \leq f(T) + \alpha f(u^*),
$$

which yields $\text{OPT} \leq 2f(T) + \alpha f(u^*)$.

3. $u^* \in O$ and and $u^*$ rejects the price in Line 11: In this case, we must have $T = A$, $c(u^*) > B - p(A)$ and hence

$$
c(O \setminus \{u^*\}) \leq p(A) \leq \frac{B \cdot f(A)}{f(\emptyset) + \alpha f(u^*)} \leq \frac{Bf(A)}{\alpha f(u^*)}
\tag{8}
$$

where the second inequality is due to Lemma 1. Note that all the sellers in $O \setminus \{u^*\}$ has been processed by Line 1 and each seller $u \in O \setminus \{u^*\} \setminus A$ must reject the offered price and hence satisfy $c(u) > p(u)$. So we can use this and Eqn. (8) to get

$$f(O \setminus \{u^*\}) - f(A) \leq \sum_{u \in O \setminus \{u^*\} \setminus A} f(u \mid A_u)$$

$$= \sum_{u \in O \setminus \{u^*\} \setminus A} p(u) \cdot \frac{f(A_u) + \alpha f(u^*)}{B} \leq \sum_{u \in O \setminus \{u^*\} \setminus A} c(u) \cdot \frac{f(A) + \alpha f(u^*)}{B}$$

$$\leq \frac{f(A)}{\alpha f(u^*)}(f(A) + \alpha f(u^*)) \leq \frac{1}{\alpha}f(A) + f(u^*) \tag{9}$$

Therefore, when $\alpha > 2$, we have

$$\text{OPT} \leq f(O \setminus \{u^*\}) + f(u^*) \leq (1 + \frac{1}{\alpha})f(A) + 2f(u^*) \leq 2f(T) + \alpha f(u^*) \tag{10}$$

According to Lines 6-13, the algorithm returns $T$ or $\{u^*\}$ with probability of $\frac{2}{2+\alpha}$ or $\frac{\alpha}{2+\alpha}$, respectively. So we get

$$\mathbb{E}[f(S)] = \Pr[S = T] \cdot f(T) + \Pr[S = \{u^*\}] \cdot f(u^*) \tag{11}$$

$$= \frac{2f(T)}{2+\alpha} + \frac{\alpha f(u^*)}{2+\alpha} \geq \frac{\text{OPT}}{2+\alpha} \tag{12}$$

which completes the proof. $\qquad\square$

Combining Lemmas 3-4, we immediately get:

**Theorem 1** *With $\alpha = \frac{\sqrt{13}+1}{2} \approx 2.3$, the* TripleEagleRan *mechanism can return a set $S$ satisfying* $\text{OPT} \leq \frac{\sqrt{13}+5}{2}\mathbb{E}[f(S)]$ *by offering one price to each seller in $\mathcal{N}$ and incurring at most $\mathcal{O}(n)$ value oracle queries.*

*Remark:* As mentioned in Sec. 2, we have adopted the assumption made in [9, 13] that $\forall v \in \mathcal{N}$ : $c(v) \leq B$ for the simplicity of description. This assumption can be easily removed according to the following discussion. Note that each buyer is guaranteed to be offered a price no more than $B$ in TripleEagleRan. Therefore, before running TripleEagleRan, we only need to identify a valid $u^* = \arg\max_{v:v \in \mathcal{N} \wedge c(v) \leq B} f(v)$. This can be done by first sorting all the sellers according to the non-increasing order of their values, and then offer $B$ to them one by one until seeing the first seller accepting $B$ and then that seller is clearly a valid $u^*$. After that, we can delete all the sellers rejecting $B$ and then run TripleEagleRan. Clearly, using such a method, the complexity of our mechanism on the number of value oracle queries is still $\mathcal{O}(n)$, while only one additional price query (for $u^*$) is incurred. This method can also be used for all our mechanisms described in the sequel.

## 4 Deterministic BFM for Monotone Submodular Valuations

If we directly use the LSDPricing algorithm to process all the sellers in $\mathcal{N}$, it is possible that the valuation of the sellers who accept the offered prices is no more than $f(u^*)$, while $u^*$ rejects its price and hence cannot be selected as a winner, resulting in a poor approximation ratio (especially when $f(u^*)$ is large). To address this issue, we have used randomization in TripleEagleRan. In this section, we introduce TripleEagleDet (i.e., Algoirthm 3), a deterministic BFM that circumvents the aforementioned issue using an idea of "two-phase pricing", as elaborated in the following.

In the first pricing phase (Lines 2-6), Algorithm 3 constructs a set $K$ by using the pricing rule in Line 3 to processes the sellers in $\mathcal{N} \setminus \{u^*\}$; this pricing rule is a relaxation of the pricing rule in the LSDPricing algorithm, because the valuation of the already processed sellers is not used to lower the price for the next seller to be processed.

The first pricing phase stops as soon as $f(K) \geq f(u^*)$ (Line 6). If $f(K)$ is still less than $f(u^*)$ when all the sellers in $\mathcal{N} \setminus \{u^*\}$ have been processed in the first pricing phase, then Algoirthm 3 sets the winner set $S = \{u^*\}$ (Line 8). In the case that the first pricing phase generates a set $K$ satisfying $f(K) \geq f(u^*)$, Algorithm 3 enters the second pricing phase in Line 10, where we use LSDPricing to process all the sellers not processed in the first phase (including $u^*$). Intuitively, since $f(K) \geq f(u^*)$

---
**Algorithm 3:** TripleEagleDet($\alpha$)
---
**1** $K \leftarrow \emptyset; C \leftarrow \mathcal{N};$
**2** **foreach** $u \in \mathcal{N} \setminus \{u^*\}$ **do**
**3**     $p(u) \leftarrow \frac{Bf(u|K)}{\alpha f(u^*)};$ Show $u$ the price $p(u);$
**4**     **if** $u$ *accepts* $p(u)$ **then** $K \leftarrow K \cup \{u\}$ ;
**5**     $C \leftarrow C \setminus \{u\};$
**6**     **if** $f(K) \geq f(u^*)$ **then break**;
**7** **if** $f(K) < f(u^*)$ **then**
**8**     $p(u^*) \leftarrow B; \ S \leftarrow \{u^*\};$
**9** **else**
**10**     $A \leftarrow \mathsf{LSDPricing}(K, C, \alpha f(u^*));$
**11**     $S \leftarrow$ the largest suffix of $A$ satisfying $\sum_{u \in S} p(u) \leq B;$
**12** **return** $S$
---

and $K$ is a prefix of $A$ in the second pricing phase (Line 10), we can regard $K$ as a substitute of $u^*$ and hence no longer need to worry about the issue explained in the beginning of this section.

We analyze the approximation ratio of Algorithm 3 by considering two cases. In the first case, we have $f(K) < f(u^*)$ or $p(A) \leq B$, which implies that $S = \{u^*\}$ or $S = A$ due to Line 8 and Line 11. In this case, we can directly use the two-phase pricing rules to bound the total valuation of the sellers in $O$ who rejected the offered prices and hence get Lemma 5:

**Lemma 5** *When Algorithm 3 finishes, if $f(K) < f(u^*)$ or $p(A) \leq B$, then we have* OPT $\leq$ $(2 + \alpha)f(S)$.

In the second case (i.e., $f(K) \geq f(u^*)$ and $p(A) > B$), we must have that $S$ is generated by Line 11 and $S \subset A$. In this case, bounding the performance of Algorithm 3 is more complicated and is given by Lemma 6:

**Lemma 6** *Suppose that $\alpha \in (2, \frac{\sqrt{17}+1}{2})$. When Algorithm 3 finishes, if $f(K) \geq f(u^*)$ and $p(A) > B$, then we have*

$$\text{OPT} \leq \max\left(3 + \frac{3\alpha + 4}{\alpha^2 + 2\alpha}, 2 + \frac{\alpha(2 + \alpha)}{\alpha^2 + \alpha - 4}, 2 + \frac{\alpha(4 + \alpha)}{\alpha^2 + \alpha - 2}\right) \cdot f(S) \tag{13}$$

We roughly explain the idea for proving Lemma 6 as follows. Consider the set $A$ generated by Line 10 (recall that $K$ is prefix of $A$). If the smallest suffix $H$ of $A$ satisfying $p(H) > B$ does not intersect $K$, then we can use similar reasoning as Lemma 3 to prove OPT $\leq \left(3 + \frac{3\alpha+4}{\alpha^2+2\alpha}\right) f(S)$. Under the other case (i.e., $H \cap K \neq \emptyset$), $H$ must have a prefix generated using the relaxed pricing rule of Line 3, which possibly degrades the valuation of $H$ and hence $S$ (note that $S$ is a suffix of $H$). Fortunately, since $f(K)$ is at most $2f(u^*)$ due to Line 6, we can prove that the approximation ratio of $f(S)$ is only slightly weaker than that in the case of $H \cap K = \emptyset$, i.e., OPT $\leq \max\left(2 + \frac{\alpha(2+\alpha)}{\alpha^2+\alpha-4}, 2 + \frac{\alpha(4+\alpha)}{\alpha^2+\alpha-2}\right) \cdot$ $f(S)$. Combining Lemmas 5-6, we can immediately get the performance bounds of TripleEagleDet, as shown by Theorem 2, which reveals that TripleEagleDet has a better approximation ratio than the deterministic BFM with a provable ratio of 4.75 in [9]. Note that Balkanski et al. [9] show that their BFM cannot achieve a ratio better than 4.5 even if their analysis for the 4.75-ratio is not tight.

**Theorem 2** *With $\alpha = \sqrt{6} \approx 2.45$, the* TripleEagleDet *mechanism can return a set $S$ satisfying* OPT $\leq (2 + \sqrt{6})f(S)$ *by offering one price to each seller in $\mathcal{N}$ and incurring $2n$ value oracle queries.*

## 5   BFM for Non-Monotone Submodular Valuations

In this section, we extend our TripleEagleRan algorithm to handle a non-monotone submodular valuation function $f(\cdot)$, as shown by the TripleEagleNm algorithm (i.e., Algorithm 4).

**Algorithm 4:** TripleEagleNm($\alpha$)

1   $A_1 \leftarrow \emptyset; A_2 \leftarrow \emptyset$;
2   **foreach** $u \in \mathcal{N} \setminus \{u^*\}$ **do**
3      $j \leftarrow \arg\max_{i \in \{1,2\}} f(u \mid A_i)$;
4      $p(u) \leftarrow \frac{B \cdot f(u \mid A_j)}{f(A_j) + \alpha f(u^*)}$; Show $u$ the price $p(u)$;
5      **if** $u$ *accepts* $p(u)$ **then**
6         $A_j \leftarrow A_j \cup \{u\}$;

7   **if** $\max\{f(A_1), f(A_2)\} \geq f(u^*)$ **then**
8      Let $u = u^*$ and process $u$ using Lines 3-6;
9      $A^* \leftarrow \arg\max_{X \in \{A_1, A_2\}} f(X)$;
10     $S \leftarrow$ the largest suffix of $A^*$ satisfying $\sum_{u \in S} p(u) \leq B$;
11   **else**
12     Sample a random number $Z$ from the uniform distribution $\mathcal{U}[0, 1]$;
13     **if** $Z \leq \frac{\alpha}{3+\alpha}$ **then**
14       $p(u^*) \leftarrow B$; $S \leftarrow \{u^*\}$;
15     **else**
16       $q \leftarrow \arg\max_{i \in \{1,2\}} f(u^* \mid A_i)$;
17       Set $p(u^*) = B - p(A_q)$ and show $u^*$ the price $p(u^*)$;
18       **if** $f(u^* \mid A_q) \geq 0$ *and* $u^*$ *accepts* $p(u^*)$ **then** $A_q \leftarrow A_q \cup \{u^*\}$ ;
19       $T \leftarrow \arg\max_{X \in \{A_1, A_2\}} f(X)$; $S \leftarrow T$;

20   **return** $S$

The TripleEagleNm algorithm is similar in spirit to the TripleEagleRan algorithm execept that it maintains two sets of candidate winners: $A_1$ and $A_2$. In Lines 1-6 of Algorithm 4, each seller $u \in \mathcal{N} \setminus \{u^*\}$ is greedily routed to $A_j(j \in \{1, 2\})$ such that $f(u \mid A_j)$ is maximized, and then $u$ is offered a price generated using the same rule as Line 2 of Algorithm 2. After all the sellers in $\mathcal{N} \setminus \{u^*\}$ are processed, Lines 12-19 of Algorithm 4 adopt the operations similar to those in Lines 6-13 of Algorithm 1, except that $u^*$ is offered a "best-effort" price of $B - p(A_q)$ after it is greedily routed to $A_q$ by Line 16. We show the performance bounds of Algorithm 4 in Theorem 3:

**Theorem 3** *With $\alpha = 3$, the* TripleEagleNm *mechanism can return a set $S$ satisfying* OPT $\leq 12\mathbb{E}[f(S)]$ *by offering one price to each seller in $\mathcal{N}$ and incurring at most $3n$ value oracle queries.*

## 6   Performance Evaluation

We use the influence maximization application mentioned in Sec. 1 to compare the following BFMs: (1) Our TripleEagleRan (TER) and TripleEagleDet (TED) algorithms, (2) The Iterative-Pruning (IP) algorithm in [9] with a ratio of 4.75, and (3) The Random-TM (RTM) algorithm in [24] with a ratio of 5. All the algorithms are implemented using C++ and are run on a Linux server with Intel Xeon Gold 6126 @ 2.60GHz CPU and 128GB memory. Each implemented randomized algorithm is independently executed 50 times, and the average result is reported. We use three real social network datasets: (1) Flixster [5] with 28,843 nodes and 272,786 edges; (2) Epinions [29] with 75,879 nodes and 508,837 edges; and (3) Slashdot [29] with 82,168 nodes 948,464 edges. We adopt the well-known *Independence Cascade* (IC) model [27] for the influence spread function $f(\cdot)$, and follow [14] to set the activation probability $p_{u,v}$ of each edge $(u, v)$ to $1/N_{in}(v)$, where $N_{in}(v)$ denotes the set of in-neighbors of $v$. The cost $c(u)$ of each node is generated uniformly at random from the interval $[0, 1]$. Since evaluating $f(\cdot)$ is known to be #P-hard [14], we adopt the approach in [11] to generate one million random *reverse-reachable sets* for approximately evaluating $f(\cdot)$. It is known that the influence spread function through such an approximation is still monotone and submodular [11].

The experimental results in Fig. 1 reveal that, for all three datasets, our BFMs outperform the baselines on utility (i.e., the valuation of returned solution). Specifically, TER and TED have similar utilities that outperform RTM (resp. IP) with the performance gain ranging from 55%-122% (resp. 4%-14%). Moreover, TER and TED incur much fewer value oracle queries than RTM (resp. IP) in more than one (resp. three to four) orders of magnitude. It can also be seen that, although TER and RTM are

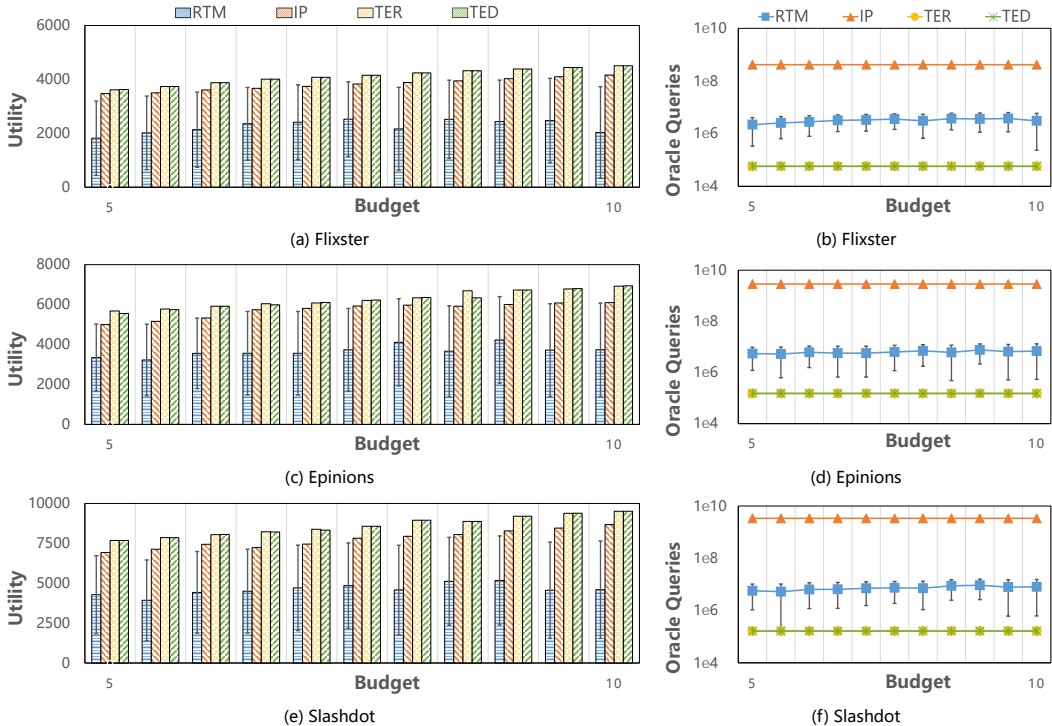

Figure 1: Comparing the BFMs for the influence maximization application

both randomized mechanisms, RTM incurs large variations while TER is almost deterministic in Fig. 1. This can be explained by the fact that RTM always returns a random solution by its design, while the randomization of TER is only triggered in some special cases (see Sec. 3). Finally, we have also compared our BFMs with the existing ones using a crowdsourcing application similar to that in [8, 16, 20, 22], where non-monotone submodular valuations are also considered. The experimental results are qualitatively similar to Fig. 1 and can be found in the Appendix due to the page limits.

# 7 Conclusion

We have proposed TripleEagle, a novel algorithmic framework for designing budget feasible mechanisms with submodular valuations, based on which we have presented several simple, fast, effective and practical BFMs that achieve obvious strategyproofness, low complexity, and better approximation ratios than the-state-of-the-art studies. The efficiency and effectiveness of our approach has been strongly corroborated by our experiments on influence maximization and crowdsourcing.

# Acknowledgements

Kai Han's work is partially supported by the National Natural Science Foundation of China (NSFC) under Grant No. 62172384.

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

# Appendix

## A  Proof of Lemma 1

**Proof**:    It is evident that $f(X) \geq \sum_{v \in X} f(v \mid A_v) = f(A_{u_s} \cup X) - f(A_{u_s})$ due to submodularity. Besides, for each $v \in X$, we have $f(A_v) \geq f(A_{u_s})$ and $c(v) \leq p(v)$ (because $v$ accepts the offered price), so we get

$$c(v) \leq p(v) = \frac{Bf(v \mid A_v)}{f(A_v) + \alpha f(u^*)} \leq \frac{Bf(v \mid A_v)}{f(A_{u_s}) + \alpha f(u^*)}$$

and hence

$$\sum_{v \in X} f(v \mid A_v) \geq \sum_{u \in X} p(v) \cdot \frac{f(A_{u_s}) + \alpha f(u^*)}{B} = p(X) \cdot \frac{f(A_{u_s}) + \alpha f(u^*)}{B}$$

which completes the proof. $\qquad\square$

## B  Proof of Lemma 2

**Proof**:    After Line 3 is executed, all the elements in $\mathcal{N}$ must have been processed by the LSDPricing algorithm. As each element in $O \setminus A$ rejects the offered price, we have

$$\forall u \in O \setminus A : c(u) > p(u) = \frac{Bf(u \mid A_u)}{f(A_u) + \alpha f(u^*)} \geq \frac{Bf(u \mid A_u)}{f(A) + \alpha f(u^*)} \tag{14}$$

Using this and submodularity, we get

$$\begin{aligned}
\text{OPT} - f(A) &\leq \sum_{u \in O \setminus A} f(u \mid A) \leq \sum_{u \in O \setminus A} f(u \mid A_u) \\
&\leq \sum_{u \in O \setminus A} c(u) \cdot \frac{f(A) + \alpha f(u^*)}{B} \leq f(A) + \alpha f(u^*) \tag{15}
\end{aligned}$$

The lemma then follows by re-arranging the above inequality. $\qquad\square$

## C  Proof of Lemma 5

**Proof**:    Let $O_1, O_2$ be the set of sellers in $O$ who reject the offered prices in Line 4 of Algorithm 3 and in the LSDPricing procedure (Line 10 of Algorithm 3), respectively. Let $K_u$ denotes the set of elements already in $K$ when $u$ is offered a price in Line 3. Under the case of $f(K) < f(u^*)$, we must have $S = \{u^*\}$ (due to Line 8), $O_1 = O \setminus \{u^*\} \setminus K$ and

$$f(O \setminus \{u^*\}) - f(K) \leq \sum_{u \in O_1} f(u \mid K_u) = \sum_{u \in O_1} p(u) \cdot \frac{\alpha f(u^*)}{B} \leq \sum_{u \in O_1} c(u) \frac{\alpha f(u^*)}{B} \leq \alpha f(u^*)$$

and hence

$$\text{OPT} \leq f(K) + \alpha f(u^*) + f(u^*) \leq (2 + \alpha) f(u^*) = (2 + \alpha) f(S) \tag{16}$$

On the other hand, if $f(K) \geq f(u^*)$ and $p(A) \leq B$, then we have $O \setminus A = O_1 \cup O_2$. Note that $K$ is prefix of $A$ under this case. So we can use submodularity to get

$$\begin{aligned}
\text{OPT} - f(A) &\leq \sum_{u \in O_1} f(u \mid K_u) + \sum_{u \in O_2} f(u \mid A_u) \\
&\leq \sum_{u \in O_1} c(u) \frac{\alpha f(u^*)}{B} + \sum_{u \in O_2} c(u) \frac{f(A_u) + \alpha f(u^*)}{B} \\
&\leq c(O_1) \cdot \frac{\alpha f(u^*)}{B} + c(O_2) \cdot \frac{f(A) + \alpha f(u^*)}{B} \\
&\leq c(O) \cdot \frac{f(A) + \alpha f(u^*)}{B} \\
&\leq f(A) + \alpha f(u^*), \tag{17}
\end{aligned}$$

which yields $f(A) \leq 2f(A) + \alpha f(u^*) \leq (2 + \alpha) f(A)$ by using $f(A) \geq f(K) \geq f(u^*)$. Note that $S = A$ under this case (due to Line 11), so the lemma follows. $\qquad\square$

# D Proof of Lemma 6

**Proof**: By similar reasoning with Lemma 5, we have

$$\text{OPT} \leq 2f(A) + \alpha f(u^*) \tag{18}$$

Let $H$ be the smallest suffix of $A$ satisfying $p(H) > B$, so $H \setminus S$ contains exactly one element (denoted by $w$) which is added into $A$ right before the elements in $S$. For convenience, we define $\delta(X) = \sum_{u \in X} f(u \mid A_u)$ for any regular subset $X$ of $A$. Suppose that $f(A \setminus H) = \mu f(u^*)$, $f(K) = \lambda f(u^*)$, $\delta(K \cap H) = \tau f(u^*)$ and $p(K \cap H) = \rho B$. Clearly, we have $p(H \setminus K) \geq (1-\rho)B$, $\lambda \geq 1$ and $\tau \leq \lambda \leq 2$. We consider the following cases:

1. $K \cap H = \emptyset$: In this case, all the sellers in $H$ are selected using the pricing rule of Algorithm 2. So we can use similar reasoning as Lemma 3 to get

$$\text{OPT} \leq \left( 4 + \frac{2 - \alpha + 2/x}{x - 1/x} \right) f(S) \tag{19}$$

   where $x = \mu + \alpha$. Note that $f(K) \geq f(u^*)$ and $K \subseteq A \setminus H$, so $\mu \geq 1$ and $x \geq \alpha + 1$. The R.H.S. of Eqn (19) is no more than $4f(S)$ when $x \geq \frac{2}{\alpha-2}$ or $\alpha + 1 \geq \frac{2}{\alpha-2}$. When $\alpha + 1 < \frac{2}{\alpha-2}$ (i.e., $\alpha < \frac{\sqrt{17}+1}{2}$) and $x \in [\alpha+1, \frac{2}{\alpha-2})$, the R.H.S. of Eqn (19) is maximized when $x = \alpha + 1$. Therefore, for any $\alpha \in (2, \frac{\sqrt{17}+1}{2})$ we have

$$\text{OPT} \leq \left( 4 + \frac{\alpha - \alpha^2 + 4}{\alpha^2 + 2\alpha} \right) f(S) = \left( 3 + \frac{3\alpha + 4}{\alpha^2 + 2\alpha} \right) f(S) \tag{20}$$

2. $|K \cap H| > 1$: In this case, we must have $|K \cap S| \geq 1$ because $|H \setminus S| = 1$ by definition. According to the construction of $K$, this implies $f(K \setminus S) < f(u^*)$ and hence

$$f(A) \leq f(K \setminus S) + f(S) \leq f(u^*) + f(S) \tag{21}$$

   Note that each seller $u \in K$ satisfies $c(u) \leq p(u) = \frac{Bf(u|A_u)}{\alpha f(u^*)}$, so we have

$$\tau f(u^*) = \delta(K \cap H) \geq p(K \cap H) \frac{\alpha f(u^*)}{B} \geq \rho \alpha f(u^*) \tag{22}$$

   This implies $\rho \leq \tau/\alpha$ and hence $p(H \setminus K) \geq (1 - \tau/\alpha)B$. By similar reasoning as Lemma 1, we have

$$\delta(H \setminus K) \geq (1 - \rho)\left(f(K) + \alpha \cdot f(u^*)\right) \geq (1 - \tau/\alpha)(\lambda + \alpha) \cdot f(u^*) \tag{23}$$

   and hence

$$\begin{aligned} \delta(H) &= \delta(K \cap H) + \delta(H \setminus K) \\ &\geq [\tau + (1 - \tau/\alpha)(\lambda + \alpha)] \cdot f(u^*) = (\alpha + \lambda - \tau\lambda/\alpha)f(u^*) \\ &\geq (\alpha + \lambda - \lambda^2/\alpha)f(u^*), \end{aligned} \tag{24}$$

   where the last inequality is due to $\tau \leq \lambda$. Recall that $1 \leq \lambda \leq 2$ and $\alpha \in (2, \frac{\sqrt{17}+1}{2})$. So the R.H.S. of Eqn. (24) is minimized when $\lambda = 2$. Therefore, we get

$$f(S) \geq f(H) - f(u^*) \geq (\alpha + 1 - 4/\alpha)f(u^*) \tag{25}$$

   Combining Eqn. (18), Eqn. (21) and Eqn. (25), we get

$$\text{OPT} \leq 2f(S) + (2 + \alpha)f(u^*) \leq \left( 2 + \frac{2 + \alpha}{\alpha + 1 - 4/\alpha} \right) f(S) \tag{26}$$

3. $|K \cap H| = 1$: In this case, we must have $A = K \cup S$, $K \cap S = \emptyset$ and $H \setminus K = S$ according to the definition of $H$. Moreover, due to Eqn. (22) we have $f(u^*) \geq \delta(K \cap H) \geq \rho \alpha f(u^*)$ and hence $\rho \leq 1/\alpha$. Note that all the elements in $S$ is selected using the pricing rule of Algorithm 2. Therefore, using similar reasoning as Lemma 1, we get

$$f(S) \geq (1 - \rho)\left(f(K) + \alpha \cdot f(u^*)\right) \geq (1 - 1/\alpha)(\lambda + \alpha)f(u^*) \tag{27}$$

Recall that $\lambda \leq 2$ and $A = K \cup S$. Combining this with Eqn. (18) and Eqn. (27), we get

$$
\begin{aligned}
\text{OPT} \quad \leq \quad & 2f(A) + \alpha f(u^*) \leq 2f(K) + 2f(S) + \alpha f(u^*) \\
= \quad & 2f(S) + (2\lambda + \alpha)f(u^*) \\
\leq \quad & \left(2 + \frac{2\lambda + \alpha}{(1 - 1/\alpha)(\lambda + \alpha)}\right) f(S) \\
= \quad & \left(2 + \frac{2}{1 - 1/\alpha} - \frac{\alpha}{(1 - 1/\alpha)(\lambda + \alpha)}\right) f(S) \\
\leq \quad & \left(2 + \frac{4 + \alpha}{\alpha + 1 - 2/\alpha}\right) f(S) \quad\quad (28)
\end{aligned}
$$

The lemma then follows according to the above discussion. $\qquad\square$

## E  Proof of Theorem 3

It can be easily seen that Algorithm 4 offers one price to each seller and incurs at most $3n$ value oracle queries ($n$ queries for finding $u^*$ and $2n$ queries for building $A_1$ and $A_2$). The approximation ratio of Algorithm 4 can be proved by directly combining Lemma 7 and Lemma 8:

**Lemma 7** *If Line 10 of Algorithm 4 is executed, then we have* $\text{OPT} \leq 12f(S)$ *when* $\alpha = 3$.

**Proof**:     In this case, all the elements in $O$ must have been processed by Lines 3-6. Let $O_1$ (resp. $O_2$) denote the sellers in $O$ who reject the offered prices that are calculated using $A_1$ (resp. $A_2$) in Line 4. For convenience, we define $\delta_i(u) = f(u \mid A_{i,u})$ for any $u \in \mathcal{N}$, where $A_{i,u}$ denotes the set of elements already in $A_i$ at the moment that $u$ is processed by Algorithm 4. It can be seen that, for each $u \in A_i (i \in \{1, 2\})$, we have $\delta_i(u) \geq 0$, because otherwise we have $p(u) = \frac{B\delta_i(u)}{f(A_{i,u}) + \alpha f(u^*)} < 0$ and hence $p(u)$ should be rejected by $u$ due to $c(u) \geq 0$, which implies that $u$ is not in $A_i$; a contradiction. So we have $f(X) \leq f(A_i)$ for any prefix $X$ of $A_i$.

In the following, we consider the sets $A_1, A_2, A^*$ after Line 10 of Algorithm 4 is executed. For each $i \in \{1, 2\}$ and each $u \in O_i$, we can use similar reasoning as Lemma 1 to get

$$
\frac{f(u \mid A_i)}{c(u)} \leq \frac{\delta_i(u)}{c(u)} < \frac{f(A_{i,u}) + \alpha f(u^*)}{B} \leq \frac{f(A_i) + \alpha f(u^*)}{B} \leq \frac{f(A^*) + \alpha f(u^*)}{B} \quad\quad (29)
$$

Moreover, according to submodularity and the greedy rule of Line 3, for all $i, j \in \{1, 2\}$ and $i \neq j$ we have

$$
\sum_{u \in A_i \cap O} f(u \mid A_j) \leq \sum_{u \in A_i \cap O} \delta_j(u) \leq \sum_{u \in A_i \cap O} \delta_i(u) \leq \sum_{u \in A_i} \delta_i(u) \leq f(A_i) \leq f(A^*), \quad\quad (30)
$$

where the third inequality is due to $\delta_i(u) \geq 0$ for all $u \in A_i$. Using the above inequalities and submodularity, we get

$$
\begin{aligned}
f(O \cup A_1) - f(A_1) \leq \quad & \sum_{u \in O \cap A_2} f(u \mid A_1) + \sum_{u \in O_1 \cup O_2} f(u \mid A_1) \\
\leq \quad f(A^*) + & \sum_{u \in O_1} \delta_1(u) + \sum_{u \in O_2} \delta_1(u) \leq f(A^*) + \sum_{u \in O_1} \delta_1(u) + \sum_{u \in O_2} \delta_2(u) \\
\leq \quad f(A^*) + & [c(O_1) + c(O_2)] \cdot \frac{f(A^*) + \alpha f(u^*)}{B} \leq 2f(A^*) + \alpha f(u^*) \quad\quad (31)
\end{aligned}
$$

where the third inequality is due to the greedy rule in Line 3. Similarly, we get

$$
f(O \cup A_2) - f(A_2) \leq 2f(A^*) + \alpha f(u^*) \quad\quad (32)
$$

Combining Eqns. (31)-(32) gives us

$$
\text{OPT} \leq \sum_{i \in \{1, 2\}} f(O \cup A_i) \leq 6f(A^*) + 2\alpha f(u^*), \quad\quad (33)
$$

where the first inequality is due to submodularity and $A_1 \cap A_2 = \emptyset$. Therefore, if $p(A^*) \leq B$, then we have $S = A^*$, $f(S) \geq f(u^*)$ and hence

$$\text{OPT} \leq 6f(S) + 2\alpha f(u^*) \leq (6 + 2\alpha)f(S) \tag{34}$$

If $p(A^*) > B$, then let $H$ be the smallest suffix of $A^*$ satisfying $p(H) > B$, and we can use similar reasoning as Lemma 1 to get

$$f(S) + f(u^*) \geq f(H) \geq f(A^* \setminus H) + \alpha f(u^*) \tag{35}$$

Combining Eqn. (33) and Eqn. (35), we get

$$
\begin{aligned}
\text{OPT} \quad \leq \quad & 6f(H) + 6f(A^* \setminus H) + 2\alpha f(u^*) \\
\leq \quad & 6[f(S) + f(u^*)] + 6[f(S) + f(u^*) - \alpha f(u^*)] + 2\alpha f(u^*) \\
\leq \quad & 12f(S) + (12 - 4\alpha)f(u^*)
\end{aligned}
\tag{36}
$$

The lemma then follows by combining Eqn. (34) and Eqn. (36). □

**Lemma 8** *If Lines 12-19 of Algorithm 4 are executed, then we have* $\text{OPT} \leq 12\mathbb{E}[f(S)]$ *when* $\alpha = 3$.

**Proof**: For clarity, we use $J_1$ (resp. $J_2$) to denote the set $A_1$ (resp. $A_2$) at the moment that Algorithm 4 executes Line 12, and define $J^* = \arg\max_{X \in \{J_1, J_2\}} f(X)$. So we have $f(J^*) < f(u^*)$ due to Line 7. It can be seen from Line 18 that $u^*$ is added into $A_q$ only if (1) $\max_{i \in \{1,2\}} f(u^* \mid J_i) \geq 0$ and (2) $u^*$ accepts the price of $B - p(J_q)$. Note that the set $T$ generated by Line 19 must satisfy $f(T) \geq f(J^*)$. In the sequel, we prove that $\text{OPT} \leq 6f(T) + 2\alpha f(u^*)$ by a discussion:

1. $u^* \notin O$: In this case, we can use similar reasoning as Lemma 7 to get

$$\text{OPT} = f(O \setminus \{u^*\}) \leq 6f(J^*) + 2\alpha f(u^*) \leq 6f(T) + 2\alpha f(u^*) \tag{37}$$

2. $u^* \in O$ and $u^*$ is added into $A_q$ in Line 18; or $u^* \in O$ and $\max_{i \in \{1,2\}} f(u^* \mid J_i) < 0$: In these situations, after Line 19 is executed, we can use similar reasoning as Lemma 7 to get

$$\forall i \in \{1, 2\} : f(O \cup A_i) - f(A_i) \leq 2f(T) + \alpha f(u^*) \tag{38}$$

Note that $f(A_i) \leq f(T)$ due to Line 19. So we get

$$\text{OPT} \leq \sum_{i \in \{1,2\}} f(O \cup A_i) \leq 6f(T) + 2\alpha f(u^*) \tag{39}$$

3. $u^* \in O$ and $\max_{i \in \{1,2\}} f(u^* \mid J_i) \geq 0$ and $\{u^*\}$ rejects the price in Line 18: In this case, we must have $c(u^*) > B - p(J_q)$ and hence

$$c(O \setminus \{u^*\}) \leq p(J_q) \leq \frac{B \cdot f(J_q)}{f(\emptyset) + \alpha f(u^*)} \leq \frac{B f(J_q)}{\alpha f(u^*)} \leq \frac{B f(J^*)}{\alpha f(u^*)} \leq \frac{B}{\alpha} \tag{40}$$

where the second inequality is got by Lemma 1. Note that all the elements in $O \setminus \{u^*\}$ has been processed by Lines 3-6. Let $O_1$ (resp. $O_2$) denote the set of sellers in $O \setminus \{u^*\}$ who reject the offered prices that are calculated using $A_1$ (resp. $A_2$) in Line 4. By similar reasoning with Lemma 7, we get

$$\forall i \in \{1, 2\}, \forall u \in O_i : \frac{f(u \mid J_i)}{c(u)} \leq \frac{\delta_i(u)}{c(u)} \leq \frac{f(J^*) + \alpha f(u^*)}{B}; \tag{41}$$

$$\forall i, j \in \{1, 2\} \wedge i \neq j : \sum_{u \in O \setminus \{u^*\} \cap J_i} f(u \mid J_j) \leq f(J^*); \tag{42}$$

where $\delta_i(u)$ is defined in the same way as that in the proof of Lemma 7. Therefore, we can use Eqns. (40)-(42) to get

$$
\begin{aligned}
f(O \setminus \{u^*\} \cup J_1) - f(J_1) \quad \leq \quad & \sum_{u \in O \setminus \{u^*\} \cap J_2} f(u \mid J_1) + \sum_{u \in O_1 \cup O_2} f(u \mid J_1) \\
\leq \quad & f(J^*) + \sum_{u \in O_1} \delta_1(u) + \sum_{u \in O_2} \delta_2(u) \\
\leq \quad & f(J^*) + c(O \setminus \{u^*\}) \cdot \frac{f(J^*) + \alpha f(u^*)}{B} \\
\leq \quad & \left(1 + \frac{1}{\alpha}\right) f(J^*) + f(u^*),
\end{aligned}
\tag{43, 44}
$$

where Eqn. (43) is due to submodularity and the greedy rule in Line 4. Similarly, we get

$$f(O \setminus \{u^*\} \cup J_2) - f(J_2) \leq \left(1 + \frac{1}{\alpha}\right) f(J^*) + f(u^*) \tag{45}$$

Combining the above equations, we get

$$f(O \setminus \{u^*\}) \leq \sum_{i=1}^{2} f(O \setminus \{u^*\} \cup J_i) \leq 2\left(2 + \frac{1}{\alpha}\right) f(J^*) + 2f(u^*) \tag{46}$$

and hence

$$\text{OPT} \leq f(O \setminus \{u^*\}) + f(u^*) \leq 2\left(2 + \frac{1}{\alpha}\right) f(J^*) + 3f(u^*), \tag{47}$$

Therefore, when $\alpha \geq 1.5$, we have $\text{OPT} \leq 6f(J^*) + 2\alpha f(u^*) \leq 6f(T) + 2\alpha f(u^*)$.

According to Lines 12-19 of Algorithm 4, the algorithm returns $T$ or $\{u^*\}$ with probability of $\frac{3}{3+\alpha}$ or $\frac{\alpha}{3+\alpha}$, respectively. So we get

$$
\begin{aligned}
\mathbb{E}[f(S)] &= \Pr[S = T] \cdot f(T) + \Pr[S = \{u^*\}] \cdot f(u^*) \\
&= \frac{3f(T)}{3+\alpha} + \frac{\alpha f(u^*)}{3+\alpha} \geq \frac{\text{OPT}}{12}
\end{aligned}
\tag{48}
$$

when $\alpha = 3$. This completes the proof. □

## F    Additional Experiments

In this section, we conduct experiments using a crowdsourcing application similar to that in [8, 16, 20, 22], where each worker (seller) in $\mathcal{N}$ owns an image, and a buyer with a budget $B$ needs to crowdsource a set $S$ of representative images from the workers. Following [8, 16, 20, 22], we use the CIFAR-10 dataset [8, 16, 20, 22] containing ten thousands $32 \times 32$ color images, where each image is associated with a label denoting its category such as "Airplane", and the valuation (utility) of $S$ is calculated as

$$f(S) = \sum_{u \in \mathcal{N}} \max_{v \in S} s_{u,v} - \frac{1}{|\mathcal{N}|} \sum_{u \in S} \sum_{v \in S} s_{u,v}, \tag{49}$$

where $s_{u,v}$ denotes the similarity between any two images $u$ and $v$ and is measured by the cosine similarity of the 3,072-dimensional pixel vectors of image $u$ and image $v$. Intuitively, the first and second additive factors in Eqn. (49) represent the "coverage" and "diversity" of $S$, respectively, and it is known that such a valuation function $f(\cdot)$ is non-monotone and submodular [16]. We assume that each worker $u$ has a private cost $c(u)$ for providing its image, and set $c(u)$ to be in proportional to the standard deviation of its pixel intensities. Following the same setting as that in [22], the costs of all images are normalized with the average value of $0.1$, and the images with labels in {Airplane, Automobile, Bird} are considered. The other experimental settings are the same with those in Fig. 1.

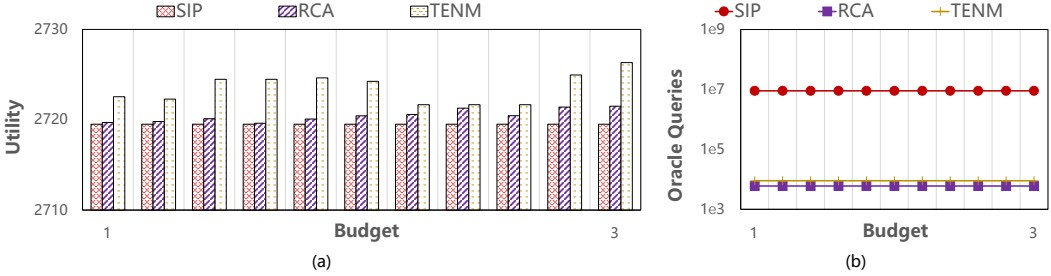

Figure 2: Comparing the BFMs for the crowdsourcing application

In Fig. 2, we compare our TripleEagleNm (TENM) algorithm with two state-of-the-art BFMs for non-monotone submodular valuations: (1) the Simultaneous-Iterative-Pruning (SIP) algorithm in

[9] with an approximation ratio of 64; and (2) the RandomClockAuction (RCA) algorithm in [22] with an approximation ratio of $(3 + \sqrt{5})^2$. The experimental results in Fig. 2 show that, our TENM algorithm consistently outperforms the other two baselines on utility, while incurring about three orders of magnitude fewer value oracle queries than the SIP algorithm. It is also noted that TENM and RCA perform similarly on the number of value oracle queries; this can be explained by the fact that RCA has a nearly-linear query complexity of $\mathcal{O}(n \log n)$ and hence is also efficient for this problem instance (but with poorer performance on utility than TENM). These experimental results demonstrate the superiorities of our approach once again for non-monotone submodular valuations.

The code of our paper can be found at:

`https://anonymous.4open.science/r/TripleEagle-4D1B/README.md`

