# OpenReview forum: "Triple Eagle: Simple, Fast and Practical Budget-Feasible Mechanisms"
_NeurIPS.cc/2023/Conference — NeurIPS 2023 poster_

### Official Review · Reviewer_LiBq · 2023-07-04

**Soundness:** 4 excellent
**Presentation:** 4 excellent
**Contribution:** 3 good
**Rating:** 7
**Confidence:** 3

**Summary:**

The paper studies the design of budget-feasible mechanisms: a buyer wants to purchase from a set of potential sellers with different production costs which are private information, and the goal is to maximize the total value subject to a budget constraint.  The main result is a new framework that achieves (1) good approximation to the first-best and (2) low pricing complexity, roughly measured by the number of questions each seller has to answer.  The authors conduct experiments that show the new methods outperform existing methods on influence maximization tasks.

**Strengths:**

The problem is very well established and practically relevant.  The paper is well written and technical ideas are clearly explained.  The paper makes solid progress for the problem by proposing a new design framework, which might trigger further progress.  The technical contribution appears to be nontrivial.

**Weaknesses:**

While I like the paper overall, one minor complaint is it doesn't say much about lower bounds.  Of course this is largely due to the intrinsic complexity of the BFM problem.

**Questions:**

(also including detailed comments here)

Line 117, "... as simple as a posted-pricing mechanism ...": maybe this will become clear later, but do you mean you offer a single price to each seller, and the seller can possibly be selected only if they accept the price?  It might help to be explicit about the connection to and difference from posted pricing early on.

Line 150, "... because otherwise we can ...": I guess this would add another pricing query to each seller?  Is this why you keep writing O(1) rather than 1?

Last paragraph in Section 6: it would be nice to explicitly comment on the tradeoff between queries and quality of solution for all 3 (families) of methods.  The impression I got is RTM makes much fewer queries at the cost of worse quality of solution, while IP returns decent solutions at the cost of much more queries.  (And then of course TER / TED has the best performance in terms of both.)

---

> ### Author Rebuttal · Authors · 2023-08-09
>
>
> **Comment\:While I like the paper overall, one minor complaint is it doesn't say much about lower bounds. Of course this is largely due to the intrinsic complexity of the BFM problem.**
>
> Response: Thanks a lot for your comments! For BFMs with additive valuation functions, Singer \[35] has proposed a lower bound of 2, and Chen et al. \[12] have proposed two improved lower bounds of 2 (for randomized BFMs) and $1+\sqrt{2}$ (for deterministic BFMs). Since submodular functions generalize additive functions, these lower bounds also apply to BFMs with submodular objectives. In Lines 34-36 of Section 1.1, we have listed the lower bounds of Chen et al. \[12]. To the best of knowledge, since the work of Chen et al. \[12] in SODA’11, no other papers have provided any further lower bounds for BFMs with submodular objectives, which remains as an interesting open problem and we will study it in future. We will add more discussions about this.
>
> **Question: Line 117, "... as simple as a posted-pricing mechanism ...": maybe this will become clear later, but do you mean you offer a single price to each seller, and the seller can possibly be selected only if they accept the price? It might help to be explicit about the connection to and difference from posted pricing early on.**
>
> Response: Nice question! You are correct. We have tried to briefly explain the differences between posted pricing and clock auctions in Lines 100-102 due to the space constraints, and we will follow your suggestion to revise these lines to make the differences clearer. Here is a detailed explanation:
>
> Suppose that each user is offered one price in a clock auction. Then the only difference between this clock auction and any posted-pricing mechanism is about how they treat any seller u who accepts the offered price: it is mandatory to select u as a winner in the posted-pricing mechanism, while u can be either a winner or a loser in the clock auction (this implies that the clock auction is allowed to use the observation of all the sellers’ behaviors in the end of the auction to decide whether u should be selected as a winner). However, from the perspective of pricing complexity, these two mechanisms are the same. That’s why we say “…virtually as simple as…”.
>
> **Question: Line 150, "... because otherwise we can ...": I guess this would add another pricing query to each seller? Is this why you keep writing O(1) rather than 1?**
>
> Response: Thanks a lot! You are right. As claimed in Lines 149-151, we have adopted the assumption made in \[9, 12] that no user’s cost is larger than B (our mechanism offers only one price to each user under this assumption). As you see, Lines 149-151 also suggest a simple method to remove this assumption by using an additional query to each seller. So we write O(1) instead of 1 for rigorousness.
>
> Actually, there is another simple method to remove the assumption in Lines 149-151 by using at most 1 additional query in total. Specifically, since each user is guaranteed to be offered a price no more than B in our main algorithm (which rules out every seller with a cost larger than B), we only need to identify a valid $v^*$ before running our algorithm. This can be done by first sorting all the sellers according to the non-increasing order of their values, and then offer B to them one by one until seeing the first seller accepting B and then that seller is clearly a valid $v^*$. After that, we delete all the sellers rejecting B and run our algorithm. Clearly,using such a method, eventually $v^*$ is queried for at most 2 times and any other seller is queried once. We have omitted this method for conciseness and would add a discussion on it if needed.
>
> **Question: Last paragraph in Section 6: it would be nice to explicitly comment on the tradeoff between queries and quality of solution for all 3 (families) of methods. The impression I got is RTM makes much fewer queries at the cost of worse quality of solution, while IP returns decent solutions at the cost of much more queries. (And then of course TER /TED has the best performance in terms of both.)**
>
> Response: Thanks a lot! You are correct. We will fully follow your suggestions to add these comments.

---

> > ### Comment · Reviewer_LiBq · 2023-08-18
> >
> > Thank you for your response!  I have no further questions.

---

> > > ### Author Response · Authors · 2023-08-19
> > >
> > > You are very welcome!

---

### Official Review · Reviewer_QWwV · 2023-07-06

**Soundness:** 4 excellent
**Presentation:** 3 good
**Contribution:** 4 excellent
**Rating:** 7
**Confidence:** 3

**Summary:**

This paper proposes a novel technique in designing budget-feasible mechanisms (BFMs), which improves the state of the art results on approximation guarantees and query complexity simultaneously. In particular:

* With monotone submodular functions, the newly proposed mechanisms improve the approximation from 4.75 to $\frac{\sqrt{13}+5}2$ (randomized BFM) and $2+\sqrt{6}$ (deterministic BFM) and improve the query complexity from $O(n^2 \log n)$ to $O(n)$;
* With non-monotone submodular functions, the newly proposed mechanisms improve the approximation from $14+6\sqrt{5}$ to 12 and improve the query complexity from $O(n \log n)$ to $O(n)$.

The key innovation of the technique is to set query prices as a decreasing function of the value of the set of sellers accepted the offered prices so far.
$$\mathsf{price}(u) = \mathsf{budget} \cdot \frac{\mathsf{marginal ~ value}(u | \mathsf{accepted ~ sellers})}{\mathsf{value}(\mathsf{accepted ~ sellers}) + \alpha \cdot \max_{s}\mathsf{value}(s)}.$$

With such adaptive price queries, each seller is queried with a price at most once. The final winners are then a subset of the sellers who accept the offers. One key advantage of this approach is that the query complexity becomes $O(n)$, and in the meanwhile the approximation guarantees are also improved with careful analysis and proper optimization of the parameter $\alpha$.

Finally, the empirical evaluation shows that the actual performance of the newly proposed mechanisms indeed dominates the benchmarks on both optimality and the number of queries.

**Strengths:**

* A significant improvement with a novel technique on a widely studied problem.
* Simultaneously improvements on both approximation guarantee and query complexity.
* Solid analysis on proving the approximation ratios.

**Weaknesses:**

* No empirical evaluation for the non-monotone submodular function case. (Addressed after author response)

**Questions:**

1. Is the current pricing formulation optimized?
2. Will querying the sellers in the order of their values be helpful?

**Limitations:**

No specific limitation.

---

> ### Author Rebuttal · Authors · 2023-08-09
>
> **Comment: No empirical evaluation for the non-monotone submodular function case.**
>
> Response: Thanks a lot for your comment! Due to the page limits, the empirical evaluation for non-monotone submodular objectives is provided in the supplemental file (Appendix F), and we have mentioned about this in Lines 312-315 in Section 6 of the main file.
>
> **Question: Is the current pricing formulation optimized?**
>
> Response: Thanks a lot for your question! We have carefully optimized the performance of our pricing method by adjusting the parameter \alpha (e.g., see Lemma 3, Lemma 6, and Theorems 1-3) to get better ratios than the existing studies. We have also listed in Lines 34-36 the currently best-known lower bounds for BFMs with submodular objectives (which were proposed by Chen et al. \[12]), i.e., 2 for randomized BFMs and $1+\sqrt{2}$ for deterministic ones. To the best of knowledge, since the work of Chen et al. \[12] in SODA’11, no other papers have provided any further lower bounds for BFMs with submodular objectives. It is interesting to make further optimizations for our algorithms to narrow the gaps between our performance ratios and the lower bounds in \[12] (or provide tighter lower bounds than \[12]), which will be our future work. We will follow your comments to provide more discussions about this.
>
> **Question: Will querying the sellers in the order of their values be helpful?**
>
> Response: Great question! Choosing order sounds a good idea. In fact, we had carefully thought about it before our submission,but it seemed that this idea can hardly help due to the following observations.In the traditional problem of submodular maximization with a knapsack constraint, good performance relies on selecting the elements strictly according to the non-increasing order of the ratios of marginal value to cost (e.g., see \[Maxim Sviridenko 2004] “A note on maximizing a submodular set function subject to a knapsack constraint”). However, in our setting the costs of the sellers are unknown and even can be falsely reported due to the sellers’ strategic behaviors, so the strict order mentioned above cannot be ensured no matter how we order the elements based on their values. Even if we use a sealed-bid auction and solicit the cost information from the sellers to achieve the strict order mentioned above (as done in \[12,21,22,35]), we have to trade off the approximation ratio for ensuring that the sellers report their true costs, and that’s why the sealed-bid auction approaches in \[12,21,22,35] all have worse approximation ratios than ours. Moreover, choosing order results in super linear complexities which could be unsuitable for large markets. Due to the above considerations, we had abandoned the ordering idea to ensure linear time complexity while still achieving better performance ratios than SOTA.

---

> > ### Comment · Reviewer_QWwV · 2023-08-21
> >
> > Thank you for your response!

---

> > > ### Author Response · Authors · 2023-08-22
> > >
> > > You are very welcome!

---

### Official Review · Reviewer_kDRB · 2023-07-06

**Soundness:** 3 good
**Presentation:** 4 excellent
**Contribution:** 4 excellent
**Rating:** 7
**Confidence:** 4

**Summary:**

The paper considers the problem of designing budget feasible mechanisms, in which the designer has a budget B, agents have private costs, and to each subset S of agents is associated a reward. The goal is to design a mechanism that incentives agents to truthfully reveal their privare costs and allows the designer to choose a subset S of agents whose total cost is at most B, and that maximizes the optimal reward of the designer.

The paper provides both deterministic and randomized mechanisms for this setting both for monotone and submodular reward functions and for non monotone and submodular reward functions. All these mechanisms improves the approximation of the chosen subset S with respect to the optimal choice. Not only, they achieve these results with a faster and "simpler to understand" algorithm, in which agent are posted a price, they may accept or refuse the price, but the actual set of selected agents will be computed only at the end among the agents that did not refuse a price.

**Strengths:**

The problem is interesting and received a lot of attention recently in our community.

The proposed algorithms improves over the previous ones.

**Weaknesses:**

None

**Questions:**

Can you stress the difference and similarities between your mechanisms and sequential posted-price mechanisms by Chawla et al., 2010?

Do you believe that a specific order of processing in algorithm 3 may help the performances of the mechanism? E.g., since f(u) are known (from the computation of v*), you may use thus information to discard the "worse" agents.

---

> ### Author Rebuttal · Authors · 2023-08-09
>
> **Question: Can you stress the difference and similarities between your mechanisms and sequential posted-price mechanisms by Chawla et al., 2010?**
>
> Response: Thanks a lot. Nice question! We guess that \[Chawla et al., 2010] refers to the STOC’10 paper “Multi-parameter Mechanism Design and Sequential Posted Pricing”. We think that this is an excellent paper, but it has studied a totally different problem. First, \[Chawla et al., 2010] considers a “forward auction” problem where a seller (without a budget) needs to maximize the revenue by selling a product to a group of buyers, while we consider a “reverse auction” problem where a buyer (with a budget) needs to buy some products from a group of sellers. Please note that forward auctions and reverse auctions are two separated lines of research in auction theory and that’s why our problem is raised in FOCS’10 \[35] after \[Chawla et al., 2010]. Second, a major difference is that \[Chawla et al., 2010] considers a “Bayesian setting” where the distributions of the preferences of the players are known, while we consider a “Prior-free” setting where there is no prior knowledge about the players. The only similarity between \[Chawla et al., 2010] and our work is that some prices are offered to players, but \[Chawla et al., 2010] use the “posted-pricing” scheme where each seller who accepts the offered price must be immediately selected as a winner, while we consider “clock auctions” where a seller is not mandatory to be selected as a winner if she/he accepts the offered price. Such a difference on pricing rule would lead to very different algorithm design and performance analysis even under the same problem.
>
> In summary, although \[Chawla et al., 2010] is a great paper, it belongs to a different line of research and their techniques cannot be applied to our problem. We will cite \[Chawla et al., 2010] and discuss about it.
>
> **Question: Do you believe that a specific order of processing in algorithm 3 may help the performances of the mechanism?E.g., since f(u) are known (from the computation of v\*), you may use thus information to discard the “worse” agents.**
>
> Response: Great question! Choosing order sounds a good idea. In fact, we had carefully thought about it before our submission,but it seemed that this idea can hardly help due to the following observations. In the traditional problem of submodular maximization with a knapsack constraint, good performance relies on selecting the elements strictly according to the non-increasing order of the ratios of marginal value to cost (e.g., see \[Maxim Sviridenko 2004] “A note on maximizing a submodular set function subject to a knapsack constraint”). However, in our setting the costs of the sellers are unknown and even can be falsely reported due to the sellers’ strategic behaviors, so the strict order mentioned above cannot be ensured no matter how we order the elements based on their values. Even if we use a sealed-bid auction and solicit the cost information from the sellers to achieve the strict order mentioned above (as done in \[12,21,22,35]), we have to trade off the approximation ratio for ensuring that the sellers report their true costs, and that’s why the sealed-bid auction approaches in \[12,21,22,35] all have worse approximation ratios than ours. Moreover, choosing order results in super linear complexities which could be unsuitable for large markets. Due to the above considerations, we had abandoned the ordering idea to ensure linear time complexity while still achieving better performance ratios than SOTA.

---

> > ### Comment · Reviewer_kDRB · 2023-08-14
> >
> > Thank you very much for the clarifying answers.

---

> > > ### Author Response · Authors · 2023-08-14
> > >
> > > You are welcome! That's our pleasure.

---

### Official Review · Reviewer_Mo9x · 2023-07-07

**Soundness:** 3 good
**Presentation:** 2 fair
**Contribution:** 2 fair
**Rating:** 7
**Confidence:** 4

**Summary:**

This paper studies the budget feasible mecchanism design, where a buyer needs to select a subset from strategic sellers with private costs , and pays them under a budget $B$. The goal is to maximize the valuation of the subset $S$ defined as $f(S)$. This paper gives both deterministic and randomized solutions for this problem, with improved approximation ratio and pricing complexity(defined as the maximum number of prices offered in the clock auction in the worst case), for monotone and non-monotone submodular functions, respectively. Finally, the paper provides an experimental evaluation of their methods against SOTA, and showed that the utility and the number of queries are better than the iterative-pruning(IP) and Random-TM(RTM) algorithm.

**Strengths:**

- The studied problem is well-motivated, and the theoretical results are solid.
- The paper has experimental results to show empirically their advantage over existing methods.
- The literature review is informative and pointed out some drawbacks of the existing works they mentioned.

**Weaknesses:**

- There is no formal problem setting, so it might be hard for reviewers from outside AGT community to understand this problem at the first sight.

Some minor comments for this paper:
- The reviewer strongly suggest to replace $f(O)$ by $\text{OPT}$.
- line 138, $2^{\mathcal{N}}$ should be $2^{|\mathcal{N}|}$.
- The description and the presentation of this algorithm is hard to understand. The reviewer suggests to add a line input with descriptions, e.g., the candidate set $C$, the accepted set $A$.
- $v^*$ can be replaced by $u^*$.

**Questions:**

Is there any concrete example of practical applications of this method.

**Limitations:**

The reviewer personally believes that the studied problem might be a little inappropriate for machine learning conferences. Theory conferences might be a better fit.

---

> ### Author Rebuttal · Authors · 2023-08-09
>
> **Comment: There is no formal problem setting, so it might be hard for reviewers from outside AGT community to understand this problem at the first sight.**
>
> Response: Thanks a lot for your comment! We have provided the formal problem setting in Section 2, including the formal problem formulations, the definitions of clock auctions and submodular functions, as well as the assumptions and notation descriptions.
>
> For a reader outside of the AGT community, our problem could also be understood as a simple pricing problem under incomplete information: we need to offer a proper price to each user with an unknown cost and finally select a group of users who accept the prices under the budget, such that the revenue of the selected users (cast by a submodular function) is maximized. Clearly, each user must behave truthfully on whether accepting the offered price or not for her/his own benefit (i.e.,truthfulness is guaranteed). By addressing this simple pricing problem, we have avoided using the other more complex forms of auctions (e.g., the sealed-bid auctions in prior studies \[4, 10, 12, 21, 22, 35]) that are harder for a reader outside the AGT community to understand.
>
> We will follow your comments to make further efforts to help the readers outside the AGT community to understand our problem.
>
> **Question: Is there any concrete example of practical applications of this method.**
>
>
> Response: Thanks a lot for your question! We have provided two concrete examples of practical applications including crowdsourcing and influence maximization in the first paragraph of Section 1, Section 6 and Appendix F of the supplementary file. These applications have also been used as representative applications of budget-feasible mechanisms in prior studies(e.g., \[4, 21, 22, 36, 37]). Specifically:&#x20;
>
> (1) In the influence maximization application shown in Section 1 and Section 6, there is a set of users (i.e.,sellers) in a social network and each of them has a cost for being selected as a "seed user".  After some users are selected as "seed users" by an advertiser with a budget for compensating the seed users, they will advertise the product to their friends and then the influence will propagate in the social network through the "word of mouth" effect. The goal of the advertiser is to select a proper set of seed users under the budget to maximize the "influence spread", i.e., the expectation of the number of users who are activated in the influence propagation, which is cast by a submodular function. This influence maximization problem was originally proposed by the celebrated paper \[25] (with 9900+ citations) and has aroused great attention during the past two decades (e.g., see \[5,15,31,32]). However, in practice the users may not report their true costs, so the advertiser is faced with the additional challenge of ensuring the truthfulness of the users, which turns the problem into an instance of the BFM problem studied in our paper.&#x20;
>
> (2) The crowdsourcing application is similar to the influence maximization application explained above, only with the difference that the "advertiser" and "seed users" in the influence maximization application are changed to the "crowdsourcing task owner" and "workers" in the crowdsourcing application, respectively, and that the target function is re-defined. In Appendix F of the supplementary file, we have elaborated on a concrete crowdsourcing example similar to that in \[4, 19, 21, 22, 37].
>
> Please note that our problem model is exactly the same as that proposed in the seminal paper \[35] and is very general (one can regard it as essentially a submodular knapsack problem plus the truthfulness requirement), so there also exist many other applications in various fields such as mobile crowdsensing,federated learning, experimental design, social advertising, vehicle sharing,data pricing, team formation, cellular traffic planning, and so on. This can be verified by checking the 300+ citations of the seminal paper of \[35] published in FOCS’10.&#x20;
>
> If given a chance, we will follow your comments to further clarify the applications of our method.
>
> **Comment: The reviewer strongly suggest to replace f(O) by OPT.**
>
> Response: Thanks. We will do as you suggested.
>
> **Comment: Line 138, $2^{\mathcal{N}}$ should be $2^{|\mathcal{N}|}$.**
>
> Response: Thanks. We use $2^{\mathcal{N}}$ to denote the power set of $\mathcal{N}$. Such a notation for power set has also been used in many other proposals (e.g., \[2]\[3]\[4]\[8]\[9]\[16]\[19]\[20]\[35]). We will revise to make it clearer.
>
> **Comment: The description and the presentation of this algorithm is hard to understand. The reviewer suggests to add a line input with descriptions, e.g., the candidate set C, the accepted set A.  $v^\star$ can be placed by $u^\star$**
>
> Response: Thanks. We will do as you suggested.
>
> **Comment: The reviewer personally believes that the studied problem might be a little inappropriate for machine learning conferences. Theory conferences might be a better fit.**
>
> Response: Thanks a lot for your comment! We agree on that our paper may also fit some theory conferences. However, the scope listed in the CFP of NeurIPS’23 includes “algorithm game theory”, which perfectly fits our paper. Moreover, since the representative applications of our paper include two social computing applications, i.e., influence maximization and crowdsourcing, which both have aroused great interests in the machine learning conferences/journals such as NeurIPS/JMLR/JAIR (e.g., \[15, 23,31, 32, 38]), we think that our paper also fits well to another scope listed in the CFP of NeurIPS’23, i.e., “Social and economic aspects of machine learning”.

---

> > ### Comment · Reviewer_Mo9x · 2023-08-15
> >
> > Thanks for the rebuttal, I've adjusted my score accordingly.

---

> > > ### Author Response · Authors · 2023-08-16
> > >
> > > Thank you very much!

---

### Decision · Program_Chairs · 2023-09-21

**Decision:**

Accept (poster)

**Comment:**

I am recommending this paper for acceptance. All four reviewers support acceptance (7,7,7,7). There was broad consensus that this paper advances the theoretical state-of-the-art, and that the experiments demonstrate that the gains are not just of theoretical nature.